# Assessment of Various Food Proteins as Structural Materials for Delivery of Hydrophobic Polyphenols Using a Novel Co-Precipitation Method

**DOI:** 10.3390/molecules28083573

**Published:** 2023-04-19

**Authors:** Ali Rashidinejad, Matthijs Nieuwkoop, Harjinder Singh, Geoffrey B. Jameson

**Affiliations:** 1Riddet Institute, Massey University, Private Bag 11 222, Palmerston North 4442, New Zealand; 2School of Natural Sciences, Massey University, Palmerston North 4442, New Zealand

**Keywords:** bioactive delivery, food-grade proteins, polyphenols, hydrophobic polyphenols, polyphenol–protein co-precipitation, functional foods

## Abstract

In this study, sodium caseinate (NaCas), soy protein isolate (SPI), and whey protein isolate (WPI) were used as structural materials for the delivery of rutin, naringenin, curcumin, hesperidin, and catechin. For each polyphenol, the protein solution was brought to alkaline pH, and then the polyphenol and trehalose (as a cryo-protectant) were added. The mixtures were later acidified, and the co-precipitated products were lyophilized. Regardless of the type of protein used, the co-precipitation method exhibited relatively high entrapment efficiency and loading capacity for all five polyphenols. Several structural changes were seen in the scanning electron micrographs of all polyphenol–protein co-precipitates. This included a significant decrease in the crystallinity of the polyphenols, which was confirmed by X-ray diffraction analysis, where amorphous structures of rutin, naringenin, curcumin, hesperidin, and catechin were revealed after the treatment. Both the dispersibility and solubility of the lyophilized powders in water were improved dramatically (in some cases, >10-fold) after the treatment, with further improvements observed in these properties for the powders containing trehalose. Depending on the chemical structure and hydrophobicity of the tested polyphenols, there were differences observed in the degree and extent of the effect of the protein on different properties of the polyphenols. Overall, the findings of this study demonstrated that NaCas, WPI, and SPI can be used for the development of an efficient delivery system for hydrophobic polyphenols, which in turn can be incorporated into various functional foods or used as supplements in the nutraceutical industry.

## 1. Introduction

Polyphenols are polyphenolic metabolites from plant sources with potential health benefits [1]. Rutin, naringenin, hesperidin, catechin, and curcumin are well-known hydrophobic polyphenols that are found in natural sources, such as buckwheat seeds, citrus peel, grapefruit, tea leaves, and turmeric, respectively. The chemical structure of these five phenolic compounds is presented in Figure 1. These polyphenols, except curcumin, share the same flavonoid C6-C3-C6 carbon framework as well as physiochemical features that express their biological effects. On a molecular level, these polyphenolic compounds possess potent antioxidant properties [2,3,4,5].

Despite these claimed beneficial health effects, the direct application of these polyphenols in food products is limited because of their hydrophobic nature and short shelf stability. Moreover, many polyphenols, which are highly unstable molecules, can undergo chemical and enzymatic degradation during processing and storage and be transformed into various reaction products, resulting in a decrease in their bioactivity [6,7].

Another major challenge is that polyphenols (including hydrophobic polyphenols) can interact with different components (e.g., protein and lipids) in the food matrix, and these interactions can often result in some undesirable changes in the sensorial and physicochemical properties of the food product, such as changes in color, flavor, and texture, as well as the decreased shelf life and chemical stability of the added polyphenols [8,9,10]. Therefore, efficient encapsulation/delivery systems for polyphenols that can overcome such challenges are required. A wide range of systems has already been tested for this purpose, e.g., liposomes, emulsions, gels, and coacervates that are made of various natural polymers, such as proteins, polysaccharides, and phospholipids [11,12,13].

Most of these encapsulation/delivery methods may improve the chemical stability of polyphenols to some extent, but to date, they are still not efficient enough to be used in the food industry. For instance:
-They most often give low encapsulation efficiency and/or loading capacity;-Some of them are not suitable for use in food formulations (e.g., use of toxic solvents);-Some use expensive or inconvenient processing methods that are difficult or inefficient to adapt and scale up in the food industry.

Additionally, the information on the clinical effects of hydrophobic polyphenols such as rutin, naringenin, hesperidin, curcumin, and catechin is very limited because of their poor bioavailability [14].

The encapsulation of bioactive compounds with proteins, such as caseins, soy proteins, and whey proteins, has been shown to be a favorable system for improving the stability of the entrapped compounds [15,16,17,18,19,20]. As proteins possess both hydrophilic and hydrophobic regions, they are considered naturally occurring block polymers in the field of bioactive compound delivery. In this study, we further explore the role of various food proteins, including sodium caseinate (NaCas), soy protein isolate (SPI), and whey protein isolate (WPI), as structural materials for the delivery of hydrophobic polyphenols (rutin, naringenin, hesperidin, curcumin, and catechin) using our established co-precipitation method.

The main challenge in delivering polyphenols is their interaction with different components in the food matrix, which can result in undesirable changes in the sensorial and physicochemical properties of the food product. Such changes may impact color, flavor, and texture in addition to decreasing the shelf life and chemical stability of the added polyphenols [8,9,10]. Most of these undesirable outcomes stem from the elevated crystallinity of added polyphenols, leading to their limited dispersibility and solubility in food. To overcome this obstacle, our objective was to design a proficient delivery system that could ameliorate these properties of hydrophobic polyphenols. This system may allow for the incorporation of polyphenols in food while reducing any negative impacts on the sensorial and physicochemical attributes of the food product.

We previously [21] developed a simple delivery system for delivering high concentrations of rutin using our novel co-precipitation method. The manufactured rutin-NaCas co-precipitates were then incorporated into various food products with no undesirable sensorial or physicochemical changes/properties [22]. This study further explores the role of different food proteins (NaCas, SPI, and WPI) as structural materials for the delivery of various hydrophobic polyphenols (rutin, naringenin, hesperidin, curcumin, and catechin) using our established co-precipitation method. Last but not least, this is the first study of its kind that has systematically investigated multiple polyphenols in multiple matrices by a common technique for easier and more valid comparisons. The three protein isolates of this research are frequently used in food and beverage products and share some similar properties. These include high protein content (all are high-quality sources of protein, with protein contents ranging from 90% to 96%), solubility (all three proteins are highly soluble in water), emulsifying properties (all can help stabilize emulsions and prevent separation of ingredients), foaming properties (all three proteins can be used to create foams in products such as whipped toppings, meringues, and baked goods), amino acid profile (while the specific amino acid profiles vary between the three proteins, all of them contain essential amino acids), and nutritional benefits (all three protein isolates are low in fat and carbohydrates). Before turning our attention to sensorial properties, we seek, therefore, to characterize the physicochemical properties, such as the solubility, dispersibility (a proxy for bioavailability), and crystallinity, of several polyphenols using an identical reproducible preparative procedure for these co-precipitates. The polyphenols chosen for this study represent a range of hydrophobicities; two are aglycone flavonoids and differ in the number of phenolic groups, and two are glycone flavonoids differing, primarily, in the position of attachment of the rhamnose glycone.

## 2. Results and Discussion

### 2.1. Powder Characteristics, Entrapment Efficiency, and Loading Capacity

Almost all lyophilized powders (either the polyphenol–protein co-precipitates or the precipitates of the control polyphenols) appeared different from their corresponding untreated polyphenols (images not shown). Additionally, the appearance of the lyophilized control proteins (i.e., NaCas, SPI, and WPI) was different than their original, untreated parental samples (i.e., the standard commercial samples that did not undergo any treatment). The co-precipitates of all three proteins (i.e., NaCas, SPI, and WPI) and polyphenols (i.e., rutin, naringenin, hesperidin, curcumin, and catechin) appeared to be different than both controls (i.e., untreated polyphenols and untreated proteins) not only in terms of their appearance but also in terms of their texture. Nonetheless, the co-precipitates still showed the distinct color of the lyophilized treated related polyphenols. These findings are in line with our previous study [21], where we used rutin as the polyphenol and NaCas as the protein.

Based on the results from HPLC analysis, almost all polyphenol content was retained within the co-precipitates, regardless of the source of protein and polyphenol. The EE ranged from about 97 to about 99%, with the LC ranging from 47 to 49%. Both the EE and LC values obtained in this study are consistent with those reported in our previous study [21], but they are much higher than the EE and EC values reported for the delivery of these polyphenols in previous publications using different methods for encapsulation [23,24,25,26].

Low LC is one of the major challenges facing the delivery of polyphenols. A low LC means that a high concentration of the wall material (protein in the case of the current study) is required to deliver a relatively low concentration of the polyphenol. This means that a high concentration of the delivery system must be incorporated into the carrier food, which, in some cases, can be impractical. The method we report in the current study yields high LC values, making it suitable for the delivery of rutin (and its aglycone derivate quercetin), naringenin (and its glycone derivate naringin), hesperidin (and its aglycone derivate hesperetin), curcumin, and catechin into food products. Several previous studies [15,16,17,18,19,20] highlighted the potential of using different food proteins as encapsulation materials for compounds and demonstrated that different proteins may be more effective depending on the specific compound being encapsulated. By considering a range of different proteins and compounds, our findings contribute to the broader understanding of how to effectively encapsulate polyphenols for their use in food products.

### 2.2. Morphology of Polyphenol–Protein Co-Precipitates

The SEM micrographs of untreated polyphenols (commercial powders) and the co-precipitates of the polyphenols with the three different proteins are shown in Figure 2. The SEM micrographs of polyphenol precipitates and polyphenol precipitates containing trehalose are also provided in the supplementary section (Appendix A). All untreated polyphenols (rutin, naringenin, hesperidin, curcumin, and catechin) presented a highly crystalline microstructure (sheet-like in most cases) (Figure 2). In some cases (e.g., naringenin and catechin), the evidence of flattened fibres/rods within the surface could also be seen. However, the density and porosity of the structural/molecular arrangement were different among various polyphenols that were tested in this experiment (Figure 2). The SEM images of different polyphenols obtained in the present work are similar to those reported in the previous studies, e.g., rutin [21], naringenin [27], hesperidin [28], curcumin [29], and catechin [30].

When polyphenols were treated in the absence of either protein (Figure 2) or trehalose (Appendix A), in the case of rutin and naringenin, for example, the rod-like structure became more prominent as the bundles of rods became more open. In some cases (e.g., rutin), the treatment also resulted in a substantial variation in the size of the particles/crystals so that some new crumbly amorphous material could also be seen (Appendix A). Based on these results, regardless of the presence of any of the proteins, the pH treatment and the subsequent precipitation and lyophilization resulted in dramatic changes in the microstructure of all five polyphenols that were tested in this experiment.

When the SEM micrographs of the co-precipitates of a specific polyphenol with different proteins were compared (Figure 2), significant differences were also observed in the crystallinity and morphology among the three co-precipitates. However, in all three cases (i.e., a specific polyphenol co-precipitated with three different proteins), although the ratios between the polyphenols and the protein isolates varied, the microstructure was more porous and much less crystalline than the untreated polyphenol. Similar to the differences observed among different polyphenols (untreated), considerable differences could also be seen among the co-precipitates of these different polyphenols. For example, the SEM microstructure of the co-precipitate of NaCas and catechin was much different from those of NaCas and the other four polyphenols (Figure 2). The same degree of difference was also seen in the case of WPI-catechin co-precipitates compared to the coprecipitates of WPI with other polyphenols.

The co-precipitates shown in Figure 2 contained trehalose (used as a cryoprotectant). To see if the addition of this disaccharide affects the properties of the final co-precipitates, we also manufactured the co-precipitates of different proteins and polyphenols in the absence of trehalose. The results showed that trehalose (2.5% *w/v* in the initial formulation) resulted in a more homogeneous structure of the polyphenol–protein co-precipitates. This confirms that the addition of this disaccharide has resulted in the rearrangement of the polyphenol–protein microstructure, which was expected from the effect of trehalose that we reported in our previous publication [21]. Trehalose is known to act as a cryoprotectant by preventing the formation of ice crystals during freeze-drying. The homogeneity of the co-precipitates is a crucial factor in their performance as delivery systems for polyphenols. Homogeneous co-precipitates with a uniform distribution of polyphenols and proteins can enhance their bioavailability (although not measured in this research) and efficacy by facilitating their controlled release and uptake in the body. Therefore, the addition of trehalose has a significant impact on the quality and predicted effectiveness of the final product.

There are some regular cubic particles seen in the SEM micrographs of the co-precipitates of almost all combinations (on the surface of particles), as well as the precipitates of different polyphenols (Figure 2). These are salt (NaCl) crystals formed due to the addition of base and acid during the pH adjustment process and subsequent lyophilization [21]. Notably, the addition of different proteins did not affect the formation of these regular cubic particles.

The effect of the protein matrix on the microstructure of a specific polyphenol could be related to the interactions between that protein and the polyphenol [8]. Although there are no systematic data available for the complexation/interaction between specific polyphenols and proteins that we tested in this study, such complexations between other polyphenols or some of these polyphenols (and some of these proteins) have been reported [18,31,32,33,34]. Some polyphenols are reported to not only bind onto different sites of proteins at low concentrations but also to covalently cross-link protein molecules when present at higher concentrations [35,36,37]. Rawel, Rohn and Kroll [37] confirmed the interactions between whey proteins and quercetin or rutin that resulted in blocking the lysine, tryptophan, and cysteine residues. These researchers [37] reported that the phenolic reactant was covalently bound to a β-lactoglobulin molecule, while fractions of high-molecular-weight protein were also detected by polyacrylamide gel electrophoresis, possibly due to the cross-linking of β-lactoglobulin with quercetin.

Keppler, Schwarz, and van der Goot [35] suggested that compared to smaller phenolic compounds (e.g., phenolic acids), the polyphenolic compounds with higher molecular weight (e.g., curcumin) are more likely to make cross-links with proteins. This is due to the presence of several aromatic rings on these molecules, meaning that there are more sites for the possible reactions to take place [35]. In a previous study [36], polyphenols were suggested as cross-linkers of protein-based products such as gelatin gels and gelatin-based coacervates for use as novel ingredients in the food industry. These researchers [36] observed that such cross-linking could lead to denser polymeric networks that prevented the possible extension of the peptide chains at the pH away from the isoelectric point. Concerning the present study, whether such interactions could occur between each polyphenol and each protein that we tested (and under the conditions of such study) is yet to be understood, and we are addressing this in our ongoing and future investigations. Additionally, whether the possible interactions between these polyphenols and proteins in such co-precipitates are reversible in the gastrointestinal digestion system is a question for future research.

### 2.3. Crystallinity of the Polyphenol–Protein Co-Precipitates

The X-ray diffraction patterns of different powders (i.e., polyphenol–protein co-precipitates) for rutin, naringenin, hesperidin, curcumin, and catechin are shown in Figure 3. Additionally, Appendix A shows the X-ray diffraction patterns of the powders of treated polyphenols (precipitates) in the absence and presence of trehalose.

In all five cases, untreated polyphenols were highly crystalline, with some being more crystalline than others (e.g., naringenin vs. rutin). The highly crystalline XRD patterns of untreated polyphenols observed in the present study (the recorded 2D diffractograms of these samples were spotty/speckled) agree with those reported in the previous studies. For example, Liu et al. [38] studied the crystallinity of quercetin, rutin, resveratrol, and hesperidin and reported similar diffractograms for these polyphenols to those we observed in the current study. Similar XRD patterns were also observed by Mauludin, Müller, and Keck [14] for rutin, by Ji, Yu, Liu, Jiang, Xu, Zhao, Hao, Qiu, Zhao, and Wu [39] for naringenin, by Ali et al. [40] for hesperidin, by Mauludin, Müller, and Keck [14] for curcumin, and by Krishnaswamy et al. [41] for catechin.

As seen in Figure 3, owing to the co-precipitation process, the crystallinity of the powders substantially decreased, and in most cases, it resulted in an almost amorphous structure. Compared to the diffractograms of the untreated polyphenols, two new peaks (at diffraction angles of about 2θ = 31° and 45°) were observed in the case of all co-precipitates. These peaks, which can be seen at the same locations in all treated samples, are consistent with our previous findings and belong to the salt (NaCl) generated during the pH adjustment of the solution and subsequent lyophilization [21]. We also observed these peaks in the case of treated samples in the absence of proteins (Appendix A). Moreover, crystalline sodium chloride is observed as regular cubic particles in the SEM micrographs of all treated samples (both polyphenol–protein co-precipitates and polyphenol precipitates) (Section 2.2, Figure 2 and Appendix A). The presence of salt crystals was expected in this experiment; because of the conditions of the treatment process, i.e., the involvement of NaOH for dissolution at alkaline pH, which was followed by precipitation at acidic pH using HCl (and followed by lyophilization). Another confirmation that these peaks (at diffraction angles of about 2θ = 31° and 45°) are associated with the added ions during the pH treatment (and precipitation process) is that these peaks were also present in the diffractograms of all the other treated polyphenol samples or polyphenol–protein co-precipitates (Figure 3).

The substantial decrease in the crystallinity of polyphenols after the co-precipitation process in the presence of three different proteins indicates that some of the big crystals in untreated polyphenols have shifted to smaller ones (e.g., nanocrystals) and/or changed to an amorphous state. This agrees with SEM micrographs that showed that the polyphenol–rutin co-precipitates had a different microstructure than their corresponding untreated form (Figure 2). These results also agree with the findings of our previous work [21,22]. We previously reported that the XRD patterns of untreated rutin were highly crystalline, with the recorded 2D diffractogram being spotty/speckled in nature. However, when rutin was treated, there was a dramatic decrease in the crystallinity (although still somewhat spotty in the 2D diffractogram). In the present study, the same effect was observed for the other four polyphenols (see Figure 3). These findings are also aligned with those reported by Mauludin, Müller, and Keck [14], who studied the kinetic solubility and dissolution velocity of rutin nanocrystals and reported a substantial increase in the solubility of lyophilized rutin crystals vs. untreated rutin. Furthermore, these results confirm our observations about the morphology of the polyphenol–protein co-precipitates and polyphenol precipitates (reported in Figure 2 and Appendix A, respectively), where the treated polyphenols in all five cases exhibited different microstructure than their untreated form.

In this experiment, we investigated the effect of the co-precipitation process on the crystallinity of the proteins used. The XRD patterns of both treated and untreated protein powders showed an amorphous XRD pattern, indicating that the proteins exist in an amorphous state regardless of the treatment. However, we did observe some sharp peaks related to salt crystals at diffraction angles of about 2θ = 31° and 45° after the proteins were treated, regardless of the protein type used. These peaks are attributed to NaCl crystals, which are formed on lyophilization of solutions that contain sodium ions (from treatment with NaOH to solubilize the polyphenols) and chloride ions (from treatment with HCl to drop the pH to create the encapsulates).

When we compared the XRD patterns of the untreated dry mix of each of the five polyphenols (mixed with each of the three proteins) with their co-precipitates, the peaks of the polyphenol–protein co-precipitates appeared to be broader, and most of the sharp peaks disappeared, with the exception of catechin with WPI. This confirms that the treatment used was, in general, able to create co-precipitates that were less crystalline, regardless of the type of protein or the type of polyphenol. This is in close alignment with the XRD patterns of untreated polyphenols and treated control polyphenols (Appendix A). Liu, Yao, Liu, and Yin [38] studied the crystallinity (XRD diffractograms) of rutin, hesperidin, and quercetin in the untreated form and their nanosuspensions after direct lyophilization. These authors reported that compared to raw polyphenols, the treated lyophilized polyphenols were substantially less crystalline.

Concerning the effect of protein structure, we previously reported that the XRD patterns of rutin and NaCas could be seen in the patterns of the mixture of both powders [21]. However, we observed that the weaker peaks of rutin were lost in part due to the broadening of the loss of crystallinity and, to some degree, could be due to the superposition of the scattering by amorphous NaCas. That means the XRD pattern of protein was in the background because there were different patterns for the samples without NaCas (treated rutin) and with NaCas (i.e., rutin–NaCas co-precipitates). Further, we speculated that the protein may limit the growth of polyphenol crystals during the process (precipitation and/or lyophilization) by making barriers between rutin crystals (due to the established role of phosphoproteins). Although the process of treating polyphenols in the absence of protein was also able to decrease the crystallinity of all five polyphenols substantially (Appendix A), such a decrease was much more evident when protein was present in the systems.

In a previous study, Lv et al. [42] reported that WPI could act as an inhibitor of precipitation/crystallization for the hydrophobic drug daidzein. These researchers [42] employed WPI with a different thermal treatment as a depressor for the crystallization of daidzein and studied its interaction mechanism with this hydrophobic compound. The results indicated the aggregation of WPI to form nanoparticles in the presence of daidzein, which significantly enhanced daidzein solubility (at least 2-fold) and stability (stable at 4 °C for at least two months) [42]. Nevertheless, in the current study, we observed that the effect of WPI on the crystallinity of different polyphenols was different. Furthermore, such an effect was different from that observed for the other two proteins used for the co-precipitation of the same polyphenol. For example, while both catechin–NaCas and catechin–SPI co-precipitates were almost amorphous, catechin–WPI co-precipitates possessed a high degree of crystallinity similar to that seen in the diffractogram of untreated catechin (Appendix A).

In the case of soy proteins, Tian, Xu, Cao, Li, Taha, Hu, and Pan [18] recently studied the interaction between pH-shifted β-conglycinin (from soy) and some polyphenols (hesperetin and hesperidin). They reported that both hydrophobic interactions and hydrogen bonds existed between the protein and polyphenols, which resulted in changes in the tertiary and secondary structures of the protein as well as the crystallinity of the manufactured complexes. Similar findings were also reported by other researchers in the case of β-lactoglobulin-naringenin nanocomplexes [17], NaCas–curcumin nanoparticles [16], naringenin-loaded chitosan–alginate nanoparticles [43], resveratrol-encapsulated SPI nanocomplexes [33], and zein–curcumin particles [34]. Mehranfar [15] explained that quercetin (the aglycone rutin derivative) could form a 1:1 complex with β-casein, possibly due to both hydrogen bonding and van der Waals interactions.

Whether such interactions can still exist in the case of rutin–NaCas co-precipitates prepared in the present study requires further investigation. Shpigelman, Shoham, Israeli-Lev, and Livney [17] reported that β-lactoglobulin could form nanosized complexes with naringenin, but there were no interactions between naringin (the glycosylated form of naringenin) and β-lactoglobulin. These authors [17] speculated that the absence of such interactions could be most likely due to the larger size and the more hydrophilic nature of naringin (compared to naringenin) that could have hindered the binding to the hydrophobic domains of the protein.

### 2.4. Dispersibility and Solubility of the Polyphenol–Protein Co-Precipitates

The particle size distribution of untreated (control) polyphenols vs. the polyphenols co-precipitated with NaCas, SPI, and WPI after dispersion in phosphate buffer (pH 7.0) is presented in Figure 4. As stated in Section 3.6, in the case of the current research, we measured dispersibility based on the decrease in the size of particles dispersed in the aqueous medium over time. The particle size distributions of treated polyphenols without any protein (i.e., precipitates) are also given in Appendix A. All untreated polyphenols were much less dispersible than their counterpart co-precipitates; differences among the untreated polyphenols probably relate more to the grinding procedure and initial crystal quality than to their hydrophobicity—hesperidin, curcumin, and, especially, catechin showed significant amounts of amorphous material in their X-ray diffraction patterns (Figure 3).

None of the untreated forms of rutin, naringenin, hesperidin, and curcumin (Figure 4) showed any significant dispersibility over 60 min, although untreated catechin (the least crystalline polyphenol) showed some dispersibility. All polyphenols showed a skewed distribution of particle sizes, with the tail extending to smaller sizes. Upon treatment in the absence of trehalose, crystallinity and particle size decreased substantially for all but one of the polyphenols (Appendix A) from >100 μm to 30–60 μm and for naringenin to ~6 μm. For hesperidin, there was only a small decrease in particle size from ~60 μm to 30 μm. In the cases of rutin, naringenin, and catechin, particle size became at least bi-modal with a significant population at ~0.1 nm in size, whereas curcumin developed a minor population with a particle size of ~160 μm. Treatment of the polyphenols with trehalose led to similar particle-size distributions as in the absence of trehalose, except for catechin, where the multi-modal distribution became a broadened mono-modal one. Treatment improved the dispersibility only of rutin, the most intrinsically soluble of polyphenols (Table 1). Treated polyphenols, with and without trehalose, all showed a substantial loss of crystallinity (Appendix A).

Co-precipitates of the polyphenols with NaCas showed poor dispersibility, with only rutin showing a redistribution of particle size over time to favor the smaller 0.1 μm particles. Initial particle sizes were similar to those of the treated polyphenols alone. On the other hand, co-precipitates with SPI and WPI all showed initial particle sizes greater than 100 μm, which is larger than the polyphenols alone. Over time, the SPI co-precipitates showed a smooth shift of their mono-modal particle-size distributions to smaller particle sizes of ~10 μm, with the exception of catechin, where the bimodal distribution shifted to smaller particle sizes and remained bimodal. For the WPI co-precipitates, smooth shifts to the mono-modal distributions to particle sizes of less than 10 μm were observed, except for rutin and naringenin, where particle sizes decreased smoothly to less than 1 μm over 60 min.

These findings agree with the observation that the polyphenol–protein co-precipitates were dramatically less crystalline than the untreated form of the corresponding polyphenols (see Figure 3, Appendix A). Correspondingly, different XRD patterns of untreated polyphenols and their treated forms reported in Section 2.3 further explain the reason for the much better dispersibility of the polyphenol–protein co-precipitates or the treated polyphenols (i.e., polyphenol precipitates). Mauludin, Müller, and Keck [14] reported that the lyophilized rutin nanocrystals were much more dispersible in water (dissolved within 15 min almost completely) than raw rutin.

To minimize the potential degradation of polyphenols (and whey proteins) under an alkaline pH, we, therefore, minimized their exposure to a maximum of 30 min. Moreover, based on our HPLC and UV spectroscopy observations, the degradation of these compounds under such conditions was minimal. In a previous study [44], we quantified that rutin dissolved in a solution maintained at pH 11.0 for 30 min underwent approximately 10% degradation.

The difference in dispersibility of different powders, as well as the differences between the dispersibility of untreated polyphenols and the polyphenols co-precipitated with different proteins, was also confirmed when we studied the solubility of these powders in the same medium (i.e., phosphate buffer) and quantified the concentration of each polyphenol using HPLC analysis. As seen in Figure 5, all co-precipitates of rutin, naringenin, hesperidin, curcumin, or catechin showed much higher solubility values (µg/mL) in the aqueous medium than their untreated counterparts. In some cases, such an improvement in the water solubility of these polyphenols reached almost 10-fold or higher (e.g., curcumin and catechin). Only for catechin was maximum solubility observed with the NaCas co-precipitate. For rutin, naringenin, and hesperidin, maximum solubility was achieved with the WPI co-precipitate. For curcumin, its solubility did not vary significantly with protein co-precipitate.

We observe that different proteins have different effects on the solubility of a specific polyphenol. Such differences may be associated with different affinities/interactions of different polyphenols for different proteins, which were discussed in Section 2.2 and Section 2.3. The general problem of poorly soluble compounds such as these polyphenols is their low solubility and slow dissolution rate. From the results obtained in the present investigation, and as a ‘rule of thumb’, we conclude that to obtain the observed improved solubility and the highest dissolution rate for rutin, naringenin, hesperidin, curcumin, and catechin, the treatment that we have applied in this study is effective. In previous work [21], we also compared the dispersibility and solubility of different powders obtained from the formulations co-precipitated in the presence of trehalose with that of the powders obtained from the formulations co-precipitated in the absence of this disaccharide. The results showed that trehalose resulted in a significant improvement in both parameters (i.e., dispersibility and solubility) not only over time but also at time 0. In the present study, this was also confirmed in the case of the dispersibility of the polyphenols precipitated in the absence of any protein and the presence or absence of trehalose (see Appendix A). Such a beneficial effect of trehalose has also been reported in other studies, such as those reported by Saleh et al. [45]. They [45] analyzed the effect of two methods (physical mixing and lyophilization) in combination with trehalose, gluconolactone, and hydroxyl propyl γ-cyclodextrin on the dissolution of bendroflumethiazide. The authors stated that the dissolution enhancements for bendroflumethiazide, which resulted from the addition of trehalose, showed a promising opportunity for other poorly soluble drugs [45].

Table 1 summarizes calculated hydrophobicities (XLogP3 3.0) and solubilities at pH 7.0 and 25 °C of polyphenols and their encapsulates, along with several reference compounds (rhamnose and a tetrahydroxybiphenyl molecule).

There are several key points to note here regarding the information presented in Table 1. First, there is no relationship between calculated hydrophobicity and solubility. Moreover, rutin and hesperidin share similar negative hydrophobicities; the rhamnose substituent endows the molecules with hydrophilicity but has very different solubilities, a difference that is maintained in the co-precipitates. The most hydrophobic species, curcumin, has a higher solubility than any of the other polyphenols except rutin. This highlights that solubility has less to do with hydrophobicity and more to do with the balance between intramolecular interactions of the crystalline state and the intermolecular interactions of the solvated state between water and the polyphenol. Second, the proportional increase in solubility for polyphenols co-precipitated with protein and trehalose is by a factor of at least 10 greater compared to their intrinsic solubility, except for the intrinsically more soluble rutin, where the increase is by only a factor of 3 to 5. Third, comparing the similarly (in)soluble aglycone flavonoid naringenin and rhamnosylated hesperidin shows that the protein matrix has a much greater effect in solubilizing hesperidin than naringenin. Fourth, the less hydroxylated naringenin shows poorer solubility than catechin, most noticeably in the NaCas co-precipitates. This difference may relate to protein–flavonoid interactions.

## 3. Materials and Methods

### 3.1. Chemicals and Reagents

Rutin, naringenin, hesperidin, curcumin, and (+)-catechin were purchased from Sigma-Aldrich (Castle Hill, NSW, Australia). According to the manufacturer, the products had a purity of 94 to 98% *w*/*w*. D-(+)-trehalose dihydrate (from *Saccharomyces cerevisiae*, ≥99%) was purchased from Sigma-Aldrich (Auckland, New Zealand). Soy protein isolate (SPI) was obtained from SUPRO (Solae, St. Louis, MO, USA), while the other two proteins (i.e., sodium caseinate (NaCas) and whey protein isolate (WPI)) were manufactured by Fonterra Co-operative Group Limited (Palmerston North, New Zealand). All other chemicals or reagents used were of analytical-reagent grade and were obtained from either Sigma-Aldrich (Auckland, New Zealand) or Thermo Fisher Scientific (Auckland, New Zealand).

### 3.2. Manufacture of the Polyphenol–Protein Co-Precipitates

In this study, a co-precipitation method was used to prepare various hydrophobic polyphenols (rutin, naringenin, hesperidin, curcumin, and catechin) entrapped within three different food proteins (sodium caseinate; NaCas, soy protein isolate; SPI, and whey protein isolate; WPI). A 10% (*w*/*v*) aqueous solution of each protein was brought to an alkaline pH (pH 11.0) using 10 M NaOH. Next, 10% (*w*/*v*) of each polyphenol was added, followed by the addition of 2.5% *w/v* trehalose, which was stirred for about 20 min to dissolve. The solution was then acidified (pH 4.6) using a 4 M HCl solution before being centrifuged at 3000× *g* at room temperature for 10 min. The supernatant was collected and stored at −18 °C for further processing to quantify the remaining (unentrapped) polyphenol in the system. The co-precipitates were lyophilized after freezing at −18 °C and milled using a coffee grinder. Precipitates of each polyphenol and each protein were also manufactured in the same way and at the same concentrations of each in the mixture to be used as controls. After drying, these precipitates were subjected to the milling process as well. We observed (high-performance liquid chromatography; see below) that the effect of alkalization on the chemical stability of these polyphenols during this short time is negligible. The products and methods described in this paper are the subjects of the PCT patent application WO2020095238A1 [22].

### 3.3. Entrapment Efficiency (EE) and Loading Capacity (LC) Determination

To measure the amount of the polyphenol entrapped inside the NaCas, SPI, and WPI precipitates (entrapment efficiency), the concentration of each polyphenol in the supernatants was determined by high-performance liquid chromatography (HPLC) following different methods (depending on the polyphenol) [13,21,46]. The HPLC instrument was equipped with a UV/visible diode array detector (Agilent Technologies, 1200 Series, Santa Clara, CA, USA). The column was a reverse-phase PrevailTM C18 with dimensions of 4.6 cm × 150 mm and 5 µm particle size (Grace Alltech, Columbia, MD, USA). The composition and conditions of the mobile phase varied based on the method used for the quantification of each polyphenol. For example, for the determination of naringenin and rutin, it consisted of acidic Milli-Q water (pH 3.50, 1% acetic acid *v*/*v*) and methanol at the volume ratio of 50:50 and a flow rate of 1.5 mL min^−1^ with a sample injection volume of 20 µL. In the case of catechin, as another example, the mobile phase consisted of 0.1% trifluoroacetic acid in Milli-Q water (pH 2.0) and methanol at the volume ratio of 75:25 with a flow rate of 1.5 mL/min. The sample injection volume for all cases was 20 µL. Different polyphenols were detected at different wavelengths and different retention times; e.g., naringenin and rutin were detected at 10.15 and 4.8 min, respectively. For the calibration of the HPLC column and quantification of each polyphenol in the samples, standard solutions of various concentrations of pure polyphenols (94 to 98%) in the mobile phase were used. To release the total fraction of the remaining polyphenols, the supernatants were disrupted in heated ethanol (70 °C) and filtered (0.45 µm; Thermo Scientific, Waltham, MA, USA) before injecting them into the HPLC column. Rutin, naringenin, curcumin, hesperidin, and catechin are soluble in ethanol at a concentration of about 4, 1, 10, 1, and 100 mg/mL, respectively. Finally, the EE of each polyphenol in the polyphenol–protein co-precipitates was calculated using the following equation:EE (%) = (C_total_ − C_sup_)/C_total_ × 100(1)
where C_total_ is the total (initial) concentration of polyphenol in the system, and C_sup_ is the polyphenol concentration in the supernatant. The LC values of the delivery system for different combinations were calculated according to the method described in our previous publication [21] and using the following equation:LC (%) = (Total polyphenol − Free polyphenol)/weight of co-precipitates × 100(2)

### 3.4. Morphology of the Co-Precipitates

An environmental scanning electron microscope (FEI Quanta 200, Eindhoven, The Netherlands) was used to study the morphology of the lyophilized powders, following a previously published method [13]. For each measurement, a small amount of the milled lyophilized (except for untreated polyphenols, which were commercial powders) sample was placed onto aluminum stubs using double-sided tape (stuck to them). After peeling off the backing, the sample was scooped onto the exposed tape, and the excess sample (if any) was subsequently puffed off. Afterward, approximately 100 nm of gold (Bal-tec SCD 050 sputter coater; Capovani Brothers Inc., Scotia, NY, USA) was used for sputter-coating the samples before viewing them under the microscope (with an accelerating voltage of 20 kV).

### 3.5. Powder X-ray Diffraction (XRD)

The XRD analysis was performed at 20.0 °C on a Rigaku RAPID image-plate detector (Rigaku, The Woodlands, TX, USA) set at 127.40 mm under the same conditions reported in our previous publication [21]. In brief, CuKα radiation (λ = 1.540562 Å), generated by a Rigaku MicroMax007 Microfocus rotating anode generator (Rigaku, USA) and focused by an Osmic-Rigaku metal multi-layer optic device (Rigaku, USA), was used. A small amount of each lyophilized milled sample was mounted in Hampton CryoLoops (Hampton Research, CA, USA). The loop was dipped in a tiny amount of Fomblin oil before mounting the sample. RAPID II software (Version 2.4.2, Rigaku, USA) was employed for the data collection, and the data were background-corrected and converted to a line profile with the 2DP programme (Version 1.0.3.4, Rigaku, USA). The data were then compared using CrystalDiffract software (Version 6.5.5, CrystalMaker Software Ltd., Oxfordshire, UK). To eliminate the effect of variability in the sample size in the cryo-loops, we scaled the data to the same rise in the background caused by the beam-stop shadow. All samples were analyzed in the 2θ angle range of 5° to 100°, with a narrow oscillation range of 5° being used for highlighting the number of crystals in the X-ray beam.

### 3.6. Dispersibility of the Co-Precipitates in Water

The dispersibility of the lyophilized powders and the untreated samples was measured according to the method from Ji et al. [39]. For this purpose, we used a Malvern Mastersizer 2000 (Malvern Instruments Ltd., Worcestershire, UK), which was equipped with a 4 mW He–Ne laser operating apparatus. To start with, about 30 mg of each powder was added to the dispersion unit containing phosphate buffer (10 mM NaH_2_PO_4_, pH 7.0), and this mixture was stirred (2000 rpm) for 60 min. The particle size distribution obscuration values of each sample were read at 2 min intervals and a wavelength of 632.8 nm. To avoid artifacts (large lumps) from the initial dispersion, the Time 0 min measurement was removed, and the data from 2 to 60 min were collected. A separate normalization was applied to the obscuration indices across the dispersion period for each sample by dividing each index by the maximum index. Finally, the volume distribution data were multiplied by the normalized obscuration index at each time point.

### 3.7. Solubility of the Co-Precipitates in Water

The solubility of the powder samples (a known amount) was measured after their addition to 10 mL of the aqueous medium that was used for the dispersibility experiment (see Section 3.6). After 24 h of constant stirring (300 rpm), the samples were centrifuged (3000× *g*, 20 °C, 10 min). The supernatant was collected and filtered using a 0.45 µm membrane (Thermo Scientific, Waltham, MA, USA) and then mixed with ethanol (1:4 *v*/*v*) to extract the amount of soluble polyphenol, which was finally quantified using the HPLC method described in Section 3.3.

### 3.8. Statistical Analysis

All samples were prepared in triplicate, and all measurements were repeated three times (except SEM and X-ray data) under a completely randomized experimental design. Mean values of data and standard deviations were calculated using Excel 2016 (Microsoft, Redmond, VA, USA), and significant differences between the treatments were evaluated using SPSS 20 Advanced Statistics (IBM, Armonk, NY, USA) at the *p* < 0.05 level.

## 4. Conclusions

Taken together, the findings of this study demonstrated that regardless of the type of protein or polyphenol tested, the polyphenol–protein co-precipitation method resulted in powders with substantially improved properties compared to their untreated counterpart polyphenols. We confirmed that using the polyphenol–protein co-precipitation method with NaCas, WPI, and SPI proteins led to high entrapment efficiency and high loading capacity, along with improved solubility and dispersibility of the polyphenols in water at pH 7.0. WPI showed the best enhancement of solubility and dispersibility of polyphenols studied compared to the other proteins. Therefore, we suggest that these proteins (i.e., NaCas, WPI, and SPI), as biocompatible and food-grade materials, are excellent candidates for the development of polyphenol–protein co-precipitates as an efficient delivery system for high concentrations of polyphenols.

The manufactured novel co-precipitates from this work have the potential to be incorporated into various foods for the manufacture of functional foods for the delivery of high doses of polyphenols, such as rutin, naringenin, hesperidin, curcumin, and catechin. Such co-precipitates may also be used as polyphenol supplements in the nutraceutical industry. The food proteins used for the co-precipitation process in this study (i.e., NaCas, SPI, and WPI) are Generally Recognized as Safe (GRAS) biopolymers with versatile structures and high nutritional value. Currently, we are carrying out further investigations to study the behaviour of these polyphenol–protein co-precipitates in various food systems, as well as in the gastrointestinal tract, in particular, the bioavailability of polyphenols.

## Figures and Tables

**Figure 1 molecules-28-03573-f001:**
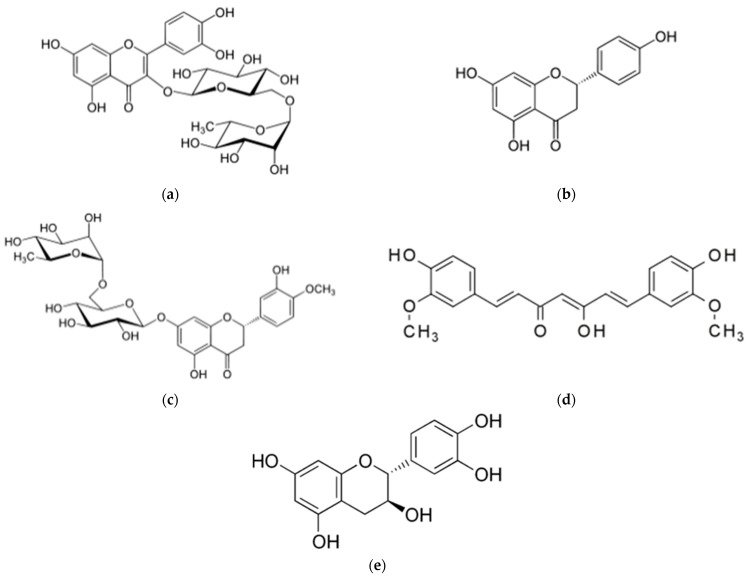
The chemical structure of the polyphenols studied in this experiment. (**a**) Rutin; (**b**) (S)-naringenin; (**c**) hesperidin; (**d**) curcumin; (**e**) (+)-catechin.

**Figure 2 molecules-28-03573-f002:**
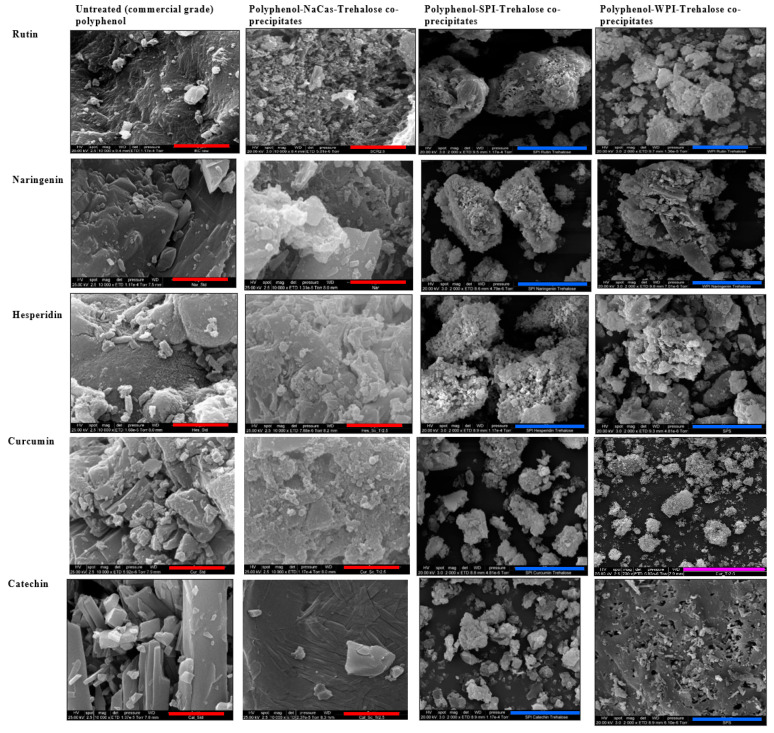
Scanning electron micrographs of the powders of untreated polyphenols and the co-precipitates of each polyphenol with different proteins in the presence of trehalose. The scale bars for each micrograph are colored red for 5 μm, blue for 30 μm and magenta for 100 μm. NaCas: sodium caseinate, SPI: soy protein isolate, and WPI: whey protein isolate.

**Figure 3 molecules-28-03573-f003:**
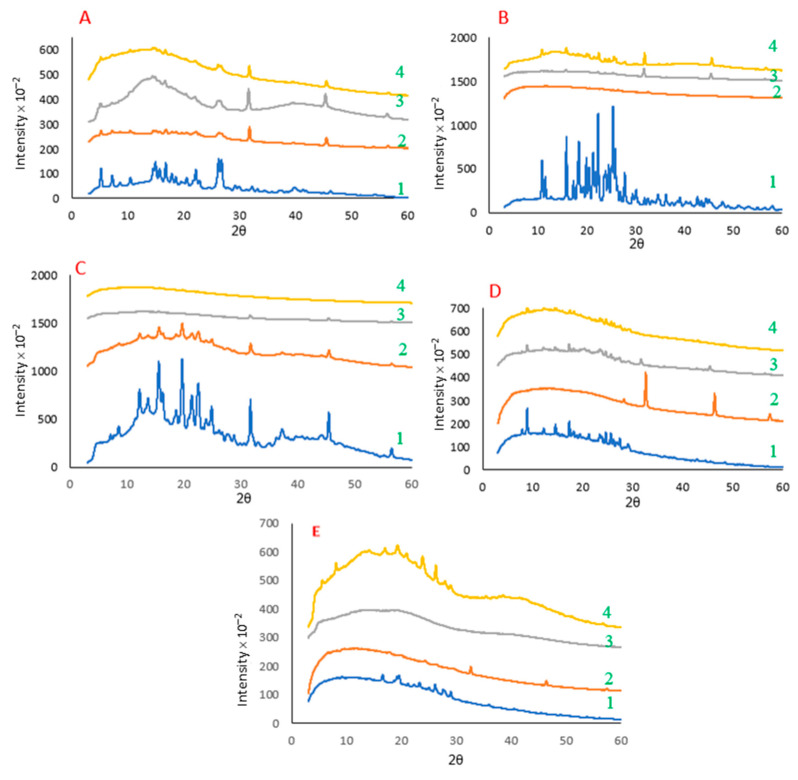
X-ray diffraction patterns of powders of untreated (commercial) polyphenols and the co-precipitates of the polyphenols with different proteins in the presence of trehalose. (**A**) Rutin; (**B**) naringenin; (**C**) hesperidin; (**D**) curcumin; (**E**) catechin. Legends from bottom to top in each graph: 1, untreated polyphenol; 2, treated in the presence of sodium caseinate (NaCas) and trehalose; 3, treated in the presence of soy protein isolate (SPI) and trehalose; 4, treated in the presence of whey protein isolate (WPI) and trehalose.

**Figure 4 molecules-28-03573-f004:**
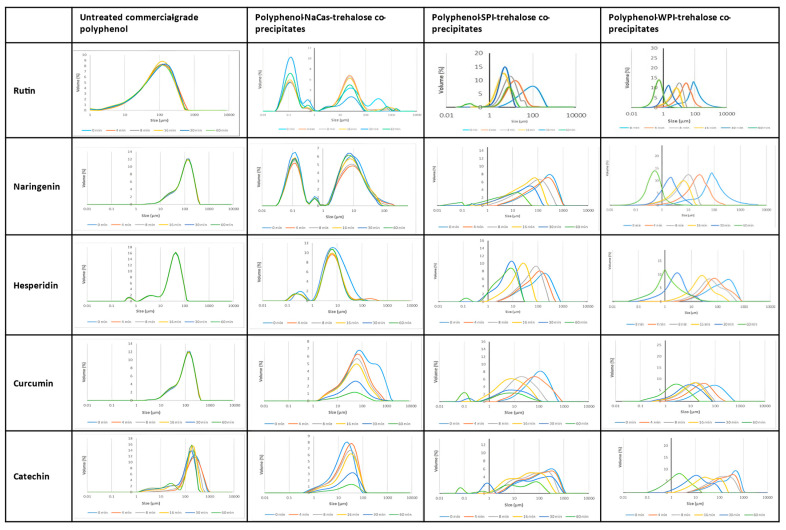
The size distribution of particles over time of the powders of untreated polyphenols and the co-precipitates of each polyphenol with different proteins in the presence of trehalose, dispersed in phosphate buffer (pH 7.0). NaCas: sodium caseinate, SPI: soy protein isolate, and WPI: whey protein isolate. Coloring is as follows: 0 (actually 2) min, cyan; 4 min orange; 8 min, grey; 16 min, yellow; 30 min, blue; 60 min, green. The distance scale extends from 0.01 nm to 10,000 nm, with the exceptions of the co-precipitate of naringenin with NaCas and trehalose where it is 0.01 nm to 100 nm, and untreated rutin where it is 1 nm to 10,000 nm.

**Figure 5 molecules-28-03573-f005:**
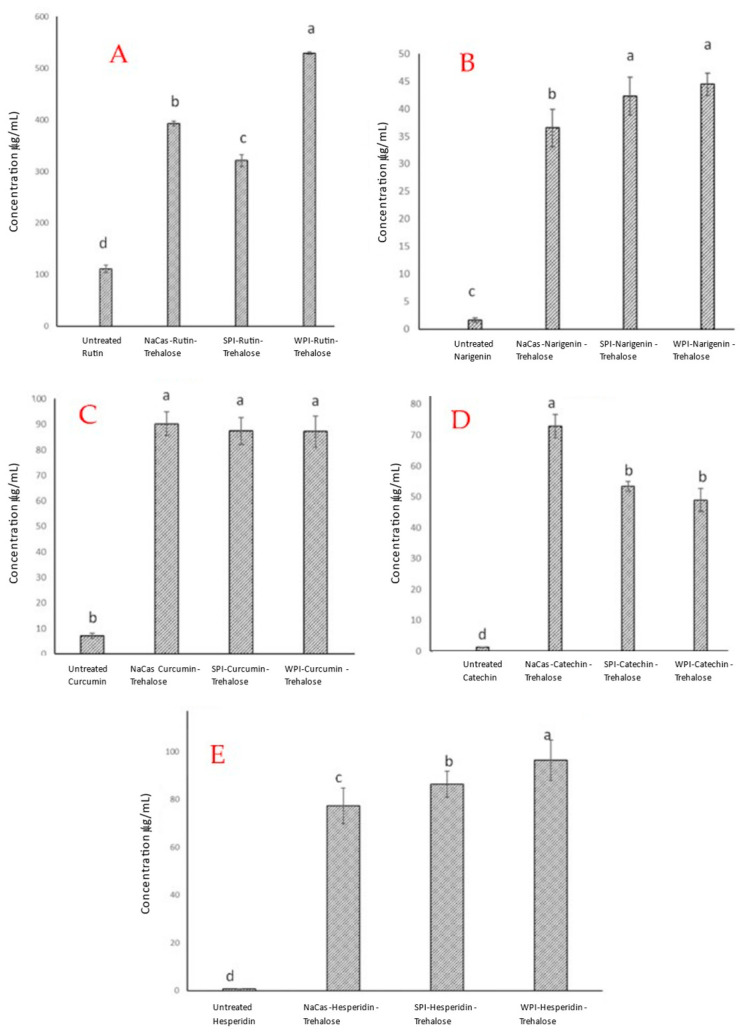
The water solubility of untreated (commercial) polyphenols and the polyphenols co-precipitated with different proteins in the presence of trehalose. NaCas: sodium caseinate, SPI: soy protein isolate, and WPI: whey protein isolate. (**A**) Rutin; (**B**) naringenin; (**C**) curcumin; (**D**) catechin; (**E**) hesperidin. In each frame the bars are ordered left to right: untreated polyphenol, NaCas-polyphenol-trehalose, SPI-polyphenol-trehalose, WPI-polyphenol-trehalose, The values in the columns with different letters are significantly different (*p* < 0.05). Error bars represent standard deviations based on three replicates.

**Table 1 molecules-28-03573-t001:** Properties of polyphenols and their co-precipitates.

Molecule	Hydrophobicity	Solubility ^a^	Solubility ^a^	Solubility ^a^	Solubility ^a^
		(Untreated)	(NaCas)	(SPI)	(WPI)
Rutin ^b,c^	−1.3 (266/610) ^d^	110	400	320	530
Naringenin	2.4 (87/272)	2	37	42	45
Curcumin	3.2 (93/368)	8	90	88	88
Catechin ^e^	0.4 (111/308)	1.5	73	53	49
Hesperidin ^b^	1.1 (234/610)	2	77	85	95
Rhamnose	−2.1 (90/164)	very high			
[1,1′-Biphenyl]-2,2′,4,4′-tetrol	2.4 (81/218)	low?			

^a^ Solubilities are given in mg/L in phosphate buffer at pH 7.0 and 25 °C. ^b^ Contains rhamnose. ^c^ Commercially available as trihydrate. ^d^ In parentheses, ratio of polar surface area (Å^3^) to molecular weight (Da)^. e^ Commercially available as the monohydrate. NaCas: sodium caseinate; SPI: soy protein isolate; WPI: whey protein isolate.

## Data Availability

Unprocessed data are available from the authors.

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
