# Peer review of "Assessment of Various Food Proteins as Structural Materials for Delivery of Hydrophobic Polyphenols Using a Novel Co-Precipitation Method"

_molecules, 2023, doi:10.3390/molecules28083573_

Round 1

Reviewer 1 Report

Report on “Assessment of various food proteins as structural materials for delivery of hydrophobic polyphenols using a novel co-precipitation method”.

This manuscript, entitled "Assessment of various food proteins as structural materials for delivery of hydrophobic polyphenols using a novel co-precipitation method," reports on the use of three different protein isolates for delivery of polyphenols. In this work, most of the studies deal with the preformulation. Although the authors state a systematic study, no reference was found in the submitted work.

First, the authors posed 4 main problems to be solved: "hydrophobicity", "degradation", "sensory properties" and "loading capacity". However, none of the points of the rose support the work, since none of them was evaluated.

·      The "hydrophobicity" was evaluated under conditions that do not correspond to real conditions (phosphate buffer 7.0);

·      There is no mention of "degradation" (or even interactions) in the document, and the "Conclusions section" states that studies on this topic are ongoing or being conducted. 

·      The “sensorial properties” are not evaluated at all.

·      The "loading capacity" is used in full in the manuscript. However, the authors would like to compare systems that encapsulate the compounds with composites/matrices in whose structure the molecules are merely entrapped. Acceptability of these results would only be given if some kind of controlled delivery were shown, which is not the case in the submitted work.

Specifically:

·      The authors claim that their study is systematic, but there is no rational discussion of the similar properties of the protein isolates used. In addition, they do not mention that the ratios between the polyphenols and the protein isolates vary. Therefore, the authors should better describe what they mean by rational.

·      The authors should indicate the stability test performed under alkaline conditions for the polyphenols.

·      The authors should describe the average values of all final pH values for each dispersion/formulation prepared.

·      It is not clear whether the milling step occurs after or before freeze-drying, only that it occurs after the freezing step (lines 115-117).

·      The authors should avoid discussing crystalline structures in the SEM section and focus on them in the XRD section. One suggestion would be to use specific technical terms for each section.

·      Figures 3, 4, and 5 need to be revised (errors) and their quality (dpi) improved. Especially for Figures 3 and 5, the titles are hardly understandable due to the small size.

So this reviewer points out that experiments need to be added and the document revised to be included in Molecules.

Author Response

Reviewer 1

This manuscript, entitled "Assessment of various food proteins as structural materials for delivery of hydrophobic polyphenols using a novel co-precipitation method," reports on the use of three different protein isolates for delivery of polyphenols. In this work, most of the studies deal with the preformulation. Although the authors state a systematic study, no reference was found in the submitted work.

First, the authors posed 4 main problems to be solved: "hydrophobicity", "degradation", "sensory properties" and "loading capacity". However, none of the points of the rose support the work, since none of them was evaluated.

  • The "hydrophobicity" was evaluated under conditions that do not correspond to real conditions (phosphate buffer 7.0);
  • There is no mention of "degradation" (or even interactions) in the document, and the "Conclusions section" states that studies on this topic are ongoing or being conducted. 
  • The “sensorial properties” are not evaluated at all.
  • The "loading capacity" is used in full in the manuscript. However, the authors would like to compare systems that encapsulate the compounds with composites/matrices in whose structure the molecules are merely entrapped. Acceptability of these results would only be given if some kind of controlled delivery were shown, which is not the case in the submitted work.

Specifically:

  • The authors claim that their study is systematic, but there is no rational discussion of the similar properties of the protein isolates used. In addition, they do not mention that the ratios between the polyphenols and the protein isolates vary. Therefore, the authors should better describe what they mean by rational.
  • The authors should indicate the stability test performed under alkaline conditions for the polyphenols.
  • The authors should describe the average values of all final pH values for each dispersion/formulation prepared.
  • It is not clear whether the milling step occurs after or before freeze-drying, only that it occurs after the freezing step (lines 115-117).
  • The authors should avoid discussing crystalline structures in the SEM section and focus on them in the XRD section. One suggestion would be to use specific technical terms for each section.
  • Figures 3, 4, and 5 need to be revised (errors) and their quality (dpi) improved. Especially for Figures 3 and 5, the titles are hardly understandable due to the small size.

So this reviewer points out that experiments need to be added and the document revised to be included in Molecules.

Response:

Thank you for the thorough review of our manuscript and the valuable feedback provided. We appreciate your efforts in reviewing our manuscript, and we have taken on board your suggestions.

In brief, we have made clearer the scope of our study, encapsulation efficiency, and loading capacity, and increased referencing to relevant literature by making the following emendations. We have also rewritten and reordered passages to improve flow and clarity.

  • We acknowledge your comment that most of the studies deal with pre-formulation and that we did not provide any reference to support our claim of a systematic study. As we discussed in the introduction, one major challenge is that these polyphenols can interact with different components in the food matrix and these interactions can often result in some undesirable changes in the sensorial and physicochemical properties of the food product. We have now elaborated on such changes including color, flavor, and texture, as well as decreased shelf-life and chemical stability of the added polyphenols, and have added the corresponding references.
  • Out of the four parameters you addressed, we did measure loading capacity. Although we did not measure the other three, it is known that they can be related to the properties of polyphenols that we have measured in this research; e.g., dispersibility, solubility, and crystallinity.
  • We have included a table to show that calculated hydrophobicity has absolutely nothing to do with the solubility of polyphenols. in the absence of ‘gold-standard” water/n-octanol partitioning experiments, we have used calculate hydrophobicities. Solubility depends very much on crystal form (or complete absence thereof). These molecules are actually pretty polar, but when crystallized they become very insoluble – they much prefer to self-associate than to associate with water as solvated molecules.
  • We actually have degradation data from HPLC traces and our earlier work, and these results, showing minimal (<10% degradation at high pH) are now included in the manuscript (Lines 152-3). We previously published a comprehensive study on the effect of pH on the various properties (including stability) of one of these polyphenols (rutin), which has now been highlighted in the new version of the manuscript.
  • Since we did not directly assess ‘’bioavailability” in this experiment, we have removed this term on Line 76.
  • The study of controlled delivery is outside the scope of this manuscript, and we will consider it for our future studies.
  • Regarding your comment about a “rational” discussion of the similar properties of the protein isolates used in the manuscript, we have now included the rationale for the choice of polyphenols.
  • The average values of all final pH for each dispersion/formulation prepared were kept similar (pH 11.0 and then acidified to pH 4.6). This has now been clarified on Lines 132 and 135.
  • We apologize for the confusion regarding the milling step and have revised the manuscript to clarify that it occurs after the freeze-drying step (Lines 161-2).
  • We have also addressed your comment regarding the discussion of crystalline structures in the SEM section by clarification and replacement of the term (Lines 304-311).
  • Regarding the figures, we have revised these figures, including improving their dpi, and made the titles more understandable.

Reviewer 2 Report

Some typographical errors were found, which are indicated in yellow in the manuscript

In Figure 2, the images corresponding to Hesperidin-WPI-Trehalose co-precipitates and Curcumin--WPI-Trehalose co-precipitates are identical. Review

Improve the quality of figures 3 and 5. Homogenize format (font size, graphic size, etc.)

Line 334, SPI should be WPI

It is recommended to be more concise with the conclusions. There is talk of conclusions from line 506 (results and decision).

Author Response

Reviewer 2

  • Some typographical errors were found, which are indicated in yellow in the manuscript
  • In Figure 2, the images corresponding to Hesperidin-WPI-Trehalose co-precipitates and Curcumin--WPI-Trehalose co-precipitates are identical. Review
  • Improve the quality of figures 3 and 5. Homogenize format (font size, graphic size, etc.)
  • Line 334, SPI should be WPI
  • It is recommended to be more concise with the conclusions. There is talk of conclusions from line 506 (results and decision).

Response:

Thank you for comments and feedback on our manuscript. We appreciate the time and effort in reviewing our manuscript and providing constructive feedback.

  • We have carefully reviewed the manuscript to identify and correct all errors indicated in yellow.
  • Regarding Figure 2 and identical images for the hesperidin-WPI-trehalose co- and curcumin--WPI-trehalose co-precipitates, we have corrected this error by inserting the correct micrograph. We apologize for the confusion caused and thank the referee for spotting this error.
  • We have improved the quality of Figures 3 and 5, made consistent their formatting, and increased font and graphics size to make them more readable and understandable. Nonetheless, in Figure 3, the scale bar could not be exactly matched as the values are different. The weight of the bars in Figure 5 also could not be harmonized, due to the initial organization of data.
  • We apologize for the error on Line 334, where SPI should be WPI. We have corrected this error in the revised manuscript.
  • Finally, we appreciate your comment on the conciseness of our conclusions. We have revised the first part of the conclusion to be more concise. We have also changed the tone of language on Line 828, to not sound like part of the main conclusion.

Reviewer 3 Report

1.        In the introduction, there needs to be a greater relationship with the research topic.

2.        Page 1 line 53 and 54,  indicate which are the undesirable changes in the

physicochemical and sensory properties of the food product.

3.        It is necessary to consult a greater number of bibliographical citations to carry out a broader discussion.

4.        In the chapter on materials and methods write the methodology more clearly.

5.        Have more discussion in the results section

Author Response

Reviewer 3

  1. In the introduction, there needs to be a greater relationship with the research topic.
  2. Page 1 line 53 and 54,  indicate which are the undesirable changes in the physicochemical and sensory properties of the food product.
  3. It is necessary to consult a greater number of bibliographical citations to carry out a broader discussion.
  4. In the chapter on materials and methods write the methodology more clearly.
  5. Have more discussion in the results section

 Response:

Thank you for taking the time to review our manuscript. We appreciate the constructive comments and suggestions for improvement. We have made the following changes to address the five points raised by the referee.

  1. We have revised the introduction to make clearer the scope of the research performed here.
  2. We have revised the manuscript to provide a clearer explanation of these changes, including references (Lines 58-60).
  3. We have reviewed our references and added several new citations to provide a more comprehensive discussion.
  4. We have revised the Materials and Methods section to make it more understandable and straightforward. It is not clear what section of the methodology this comment was referring to, but we assumed it was related to Section 2.2, and accordingly have substantially revised this section.
  5. We have revised the results section (in different paragraphs) to include a more in-depth discussion of the results, which we believe will strengthen the manuscript.